# Temporal response patterns of Layer 4 rat barrel cortex neurons across various naturalistic whisker motions

**Dinh K. Tang[1], Mark B. Flegg[2], Ramesh Rajan[1]** *

**1** School of Biomedical Sciences, Monash University, Melbourne, Victoria, Australia, **2** School of Mathematics, Monash University, Melbourne, Victoria, Australia

* ramesh.rajan@monash.edu

**Data Availability Statement:** The spike-sorted single unit data underlying the findings of this study are available from https://github.com/dk-tang/layer4-temporal-patterns.

## Abstract

A central topic in neuroscience is the neural coding problem which aims to decipher how the brain signals sensory information through neural activity. Despite significant advancements in this area, the characterisation of information encoding through the precise timing of spikes in the somatosensory cortex is limited. Here, we utilised a comprehensive dataset from previous studies to identify and characterise temporal response patterns of Layer 4 neurons of the rat barrel cortex to five distinct stimuli with varying complexities: Basic, Contact, Whisking, Rough, and Smooth. A Gaussian Mixture Model (GMM) clustering analysis was applied to identify distinct temporal response patterns. We found that three stimuli (Rough, Smooth, and Contact) produced multiple temporal response patterns while Whisking and Basic stimuli exhibited a single pattern for all conditions. These patterns of neuronal responses were differentiated by the speed and strength of the responses when more than two clusters were present. Investigation into stimulus complexity indicated that stimuli with lower complexity scores (Whisking and Basic) resulted in fewer distinct response patterns, reflecting the reduced variability in the input information signal to Layer 4. In contrast, stimuli with higher complexity scores (Rough, Smooth, and Contact) produced distinct temporal response patterns, likely driven by a broader range of deflection amplitude variations and whisker direction changes. Further analysis of neuronal responses to Contact, Rough, and Smooth stimuli revealed three broad groups of temporal response patterns: phasic on-off response, prolonged on-off response, and tonic response. We speculate that these groups of temporal response patterns encode information about the velocity, acceleration, position, direction, and continuous monitoring of whisker deflection stimuli. The observed patterns contribute to the understanding of how neurons in Layer 4 of the rat barrel cortex specialise in encoding specific features of sensory stimuli and highlight the role of stimulus complexity in shaping neuronal responses.

## Introduction

The neural coding problem refers to the suite of mechanisms by which the brain processes sensory input by representing, encoding, and decoding neural activity [1]. The count and the

**Funding:** The author(s) received no specific funding for this work.

**Competing interests:** The authors have declared that no competing interests exist.

timing of action potentials, also known as spikes, both appear to be able to encode information about the stimulus in the auditory cortex [2], visual cortex [3], and somatosensory cortex [4, 5]. However, there is still debate on whether spike count or spike timing is the primary method of encoding information. Spike count is the frequency of spikes generated by a neuron in response to a stimulus within a specific time window relative to the stimulus duration. Spike count does convey some information about the stimulus but does not capture all of the information encoded by the neuronal response [4]. Conversely, spike timing refers to the precise timing of the spikes, and the patterns of temporal responses have been shown to convey additional information beyond that encoded in the spike count about the stimulus [6]. However, the commonality and differences in temporal response patterns for encoding different sensory stimuli in the barrel cortex are not fully understood.

The rat whisker system provides information about objects that cause deflections of the face whiskers (vibrissae) and is an attractive model system for studying spike trains due to its well-defined sensory input, relatively simple circuitry, and experimental manipulability. Perception of that sensory input occurs through the interpretation of neural activity in the *barrel cortex*, a specialised region of the primary somatosensory cortex (S1) devoted to encoding sensory information derived from the deflections of the face whiskers. Barrel cortex neurons are organised into distinct columns and layers and deflection of each facial vibrissae activates a unique group of S1 *barrel column* neurons, to respond with a pattern of precisely timed spikes. Cortical information processing follows a 'canonical circuit' [7], wherein whisker-derived neural information flows from thalamus primarily into the main input layer (granular Layer 4) which then transmits output through the cortical column to the upper layers (supragranular Layers 2 and 3) from whence they propagate, after further processing, to the primary output layers (infragranular Layer 5 and 6). In this paper, we restrict our investigation to Layer 4, the primary layer that receives thalamic input and therefore the first layer of the neocortical circuitry that encodes information about the stimulus.

A well-established method for studying neuronal encoding is to obtain multi-neuronal (*multi-unit*) activity via extracellular electrophysiological recording in the barrel cortex during stimulus presentation. This method relies on population encoding, and the spike counts are averaged across multiple trials to obtain a peristimulus time histogram (PSTH) that depicts the occurrence of spikes relative to the timing and duration of a sensory stimulus. In the barrel cortex, PSTH has been extensively used to characterise differences in neuronal firing rates across cortical lamina [8], various complex naturalistic stimuli [9], and specific experimental manipulation such as traumatic brain injury [10, 11] and environmental enrichment [12, 13]. PSTH has also been used to study how neurons encode information about sensory stimuli by examining the temporal patterns of responses in the auditory cortex [14] and in the somatosensory cortex [15]. Furthermore, spike timing within the rat barrel cortex architecture conveys more information about the whisker stimulus deflection than spike count alone [16]. Here, we analyse previously collected extracellular physiological recordings from Layer 4 of the barrel cortex to identify and characterise temporal response patterns to various stimuli.

In the barrel cortex, neurons have been classified on a temporal spectrum ranging from slowly adapting to rapidly adapting [17]. Rapidly-adapting or phasic neurons respond strongly to the onset and/or offset of a stimulus, followed by a rapid decline in the firing rate even if the stimulus persists. In contrast, slowly-adapting or tonic neurons typically maintain their firing rate for extended periods, often throughout the stimulus presentation. It is unclear to what extent these phasic and tonic populations of neurons in Layer 4 contribute to the information processing across different types of stimuli with varying complexities. Our analyses of temporal response patterns across different simple and complex naturalistic stimuli provide insights

into how sensory information is encoded diversely by neurons in Layer 4. Using an unsupervised clustering approach to computationally define and classify the diversity of Layer 4 neuronal responses to various stimuli, we have identified that the formation of temporal response patterns is mainly dependent on stimulus complexity and the strength of the whisker deflection.

## Materials and methods

### Neurophysiology

All experimental procedures were approved by the Monash University Standing Committee on Ethics in Animal Experimentation (approval number MARP/2015/035) and were terminal experiments done under halothane anaesthesia. The dataset used for this paper was collated from previous studies that have reported in detail the methods of data collection, spike sorting, and pre-processing of the spikes to create PSTHs [8, 10, 13, 18].

In summary, extracellular recordings were performed in 14 anaesthetised Adult male Sprague-Dawley rats aged 10–12 weeks (335–350 g) using a parylene-coated tungsten 2–4 MΩ microelectrode (FHC, ME, U.S.A) that was inserted into the barrel cortex to be normal to the cortical surface to run down a barrel column activated by the same principal whisker (PW; whisker whose deflection elicits the most number of and most precisely-timed spikes). Electrophysiological recordings reported here were obtained from neurons in Layer 4 (750–1000 $\mu$m from the cortical surface) in response to the deflection of the PW that provided the main excitatory input to neurons in that cortical column.

It should be noted that recordings from multi-unit clusters may potentially combine information from excitatory neurons (with slow spike waveforms) and inhibitory interneurons (fast spike waveforms). However, we have not attempted to differentiate between these two types of waveforms, as the filtering applied during recording and preprocessing can obscure waveform characteristics needed to reliably differentiate fast from slow spike waveforms. We also note that with the types of electrodes used, we are unlikely to record to any significant degree from inhibitory interneurons.

A sequence of principal whisker (PW) deflections was conducted using a single stimulus waveform per recording, as depicted in Fig 1 and described below:

a. *"Basic"*. A trapezoid stimulus consisting of three distinct phases: an initial onset ramp of PW deflection at varying velocities to a distance of 3.6 mm from the whisker's rest position; a hold period of 20 ms; and an offset ramp of 40 ms back to rest [9];

b. *"Contact"*. A naturalistic object contact stimulus that, over 100 ms, mimics the motion of a large facial whisker of an unrestrained rat brushing past a metal rod, reconstructed from high-speed video-tracking of awake behaving rats; [19];

c. *"Whisking"*. A naturalistic stimulus that, over 430 ms, follows a pattern of smooth rhythmic whisker protraction and retraction as occurs in unconditioned continuous whisking behaviour in a rat at rest [20];

d. *"Rough"*. A naturalistic rough-surface-discrimination stimulus that, over 61 ms, models the videographed motion of the PW being swept against an 80-grit sandpaper surface [21]; and

e. *"Smooth"*. A naturalistic smooth-surface-discrimination stimulus that, over 85 ms, models the videographed motion of a whisker being moved over a glass surface [21].

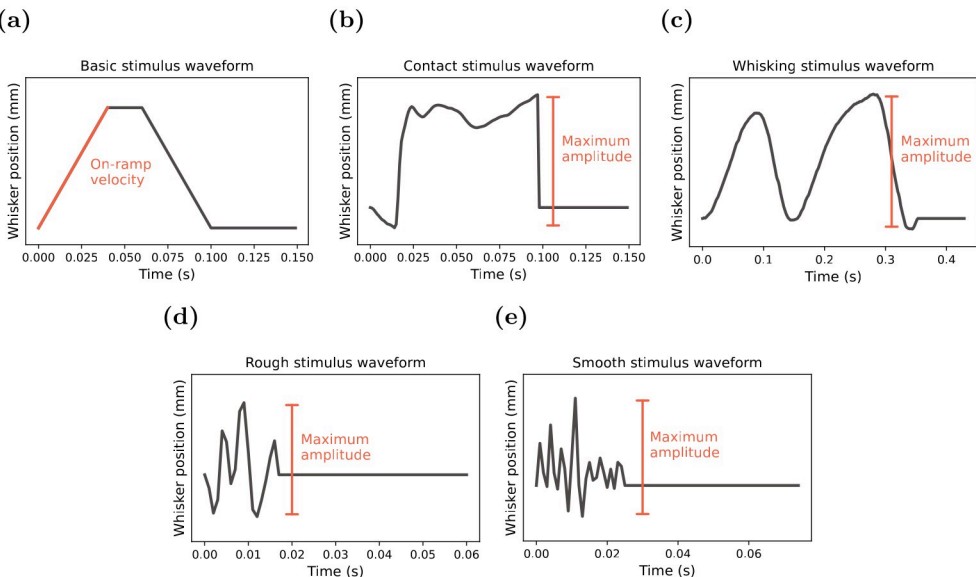

**Fig 1. Stimulus waveforms.** A plot of the PW deflection versus time for each of the stimulus waveforms studied here. These waveforms include **(a)** *"Basic"*—a trapezoid stimulus that consists of an initial onset ramp at varying velocities from the rest whisker position, a hold period, and an offset ramp back to rest; **(b)** *"Contact"*—a naturalistic object contact stimulus that mimics the motion of the whisker of an unrestrained rat brushing past a metal rod; **(c)** *"Whisking"*—a naturalistic free whisking stimulus follows a pattern of smooth rhythmic protraction and retractions in unconditioned continuous whisking behaviour; **(d)** *"Rough"*—a naturalistic rough surface discrimination stimulus that captures the motion of the PW whisker being swept against a rough surface; and **(e)** *"Smooth"*—a naturalistic smooth surface discrimination stimulus that follows the motion of the PW whisker being swept against a smooth surface.

As previously reported in Alwis (2012), the five stimuli were rated for stimulus complexity using the Normalised Length Density (NLD) method [9] which estimates Higuchi's signal fractal dimension but adjusts for short stimulation durations with less than 100 samples [22].

The Basic stimulus (a) was presented in a suite where the onset ramp velocities were varied pseudo-randomly at one of five different velocities (30, 60, 150, 250, or 400 mm/sec) while maintaining a constant maximum deflection amplitude of 3.6 mm, and a constant hold time and offset velocity. The suite of Basic stimuli at varied velocities was presented for 150–300 repetitions in a pseudo-random order. The other four stimuli (b)—(d) were each presented as a suite of the appropriate waveform with ten different standardised maximum amplitudes, ranging from 0.2 mm to 3.6 mm in increments of 0.4 mm from 0.4 mm in a pseudo-random order. Each suite of the 10 stimulus amplitudes was repeated 50 times with a one-second inter-mission between each repetition. Recorded neuronal signals were then subjected to digital amplification and a spike was detected if the signal exceeded a positive threshold of 1.5x than the average voltage. Individual single neurons (single units) were identified from the recorded multi-unit activity by highly experienced electrophysiologists who online-sorted the population of responses based on analysis of waveform features using the Cambridge Electronic Design Spike 2 software. The single units' spike responses were differentiated using spike waveform characteristics, including the width of the action potential, the rise time, the relative amplitude, and the size of the action potential overshoot. The resultant differentiated single units were termed spike-sorted single units and the sorted spiking activity for each single unit was collated in 1 ms bins with a temporal window of 200 ms before stimulus onset to 100 ms post-stimulus offset. The number of single units identified for each stimulus is reported in Table 1.

**Table 1. Summary of Layer 4 neurons across stimuli.** This table provides a breakdown of the number of animals used to identify the number of online-sorted units for each stimulus type. For each stimulus, the stimulus was applied for a fixed duration with the analysis window typically longer to capture activity after the offset of the stimulus. The stimulus complexity was measured by Normalised Length Density (NLD) as reported in Alwis (2012).

| Stimulus | Total animals | Total cells | Stimulus duration | Analysis window | Stimulus complexity |
|---|---|---|---|---|---|
| Basic | 13 | 72 | 100 ms | 150 ms | 0.000814 |
| Whisking | 7 | 69 | 340 ms | 430 ms | 0.000331 |
| Contact | 14 | 248 | 100 ms | 150 ms | 0.00117 |
| Smooth | 13 | 130 | 25 ms | 75 ms | 0.0363 |
| Rough | 13 | 136 | 17 ms | 61 ms | 0.0103 |

## Single unit spiking activity

For each recording session, single-unit spiking profiles were produced by summing the spike counts in 1 ms bins across the 50 repetitions of that stimulus (differentiated by onset ramp velocity for the Basic stimulus and by the stimulus amplitude for each of the other four stimuli) to produce a peristimulus time histogram (PSTH) for each stimulus. The temporal window of the PSTH for the single units varied between stimuli in accordance with the differences in stimulus duration. For each stimulus, the temporal window of the PSTH used for further analysis consisted of two time periods, the stimulus duration and a period of post-stimulus activity. The analysis temporal window for each stimulus is as follows: Basic—150 ms; Object—150 ms; Whisking—430 ms; Rough—61 ms; and Smooth—75 ms. The breakdown of the stimulus duration and the analysis window is reported in Table 1. In addition, we applied additional analysis to the neuronal responses to the Whisking stimulus in the first 150 ms, or 175 ms, or 225 ms of the 450 ms-long stimulus for which the GMM analysis had revealed only one distinct neuronal population response. By restricting our analysis to these shorter time periods, we could examine if our analysis would reveal finer neuronal sub-populations based on temporal dynamics that may be averaged out when using the full stimulus duration.

A visual representation of all single-unit responses is provided in the bottom panel of each stimulus in Fig 2, where each row in the raster plot corresponds to the activity of the single unit over time in response to a stimulus. Through visual examination of the raster plot, a diversity of response patterns was identified for each of these stimuli. Most notably, the Contact stimulus in panel (b) revealed the most heterogeneous responses, with some single units responding earlier or later during the stimulus onset, with some units non-responsive outside the stimulus onset and offset period. The population responses for each stimulus were constructed by averaging the spike counts of each single unit PSTH in the 1 ms time bins seen in the top panels of each stimulus of Fig 2.

## Clustering analysis

To identify and quantify the temporal patterns of single-unit responses, a Gaussian Mixture model (GMM) was deployed to group spike-sorted single-units into clusters based on the statistical properties of the temporal response profiles. The GMM is a probabilistic model that is often used to cluster data based on the assumption that the data comes from a mixture of $k$ multiple Gaussian distributions with independent means, variances, and weights. The GMM for an empirical distribution is defined as:

$$p(x) = \sum_{i=1}^{k} \alpha_i \cdot N(x|\mu_i, \sigma_i)$$

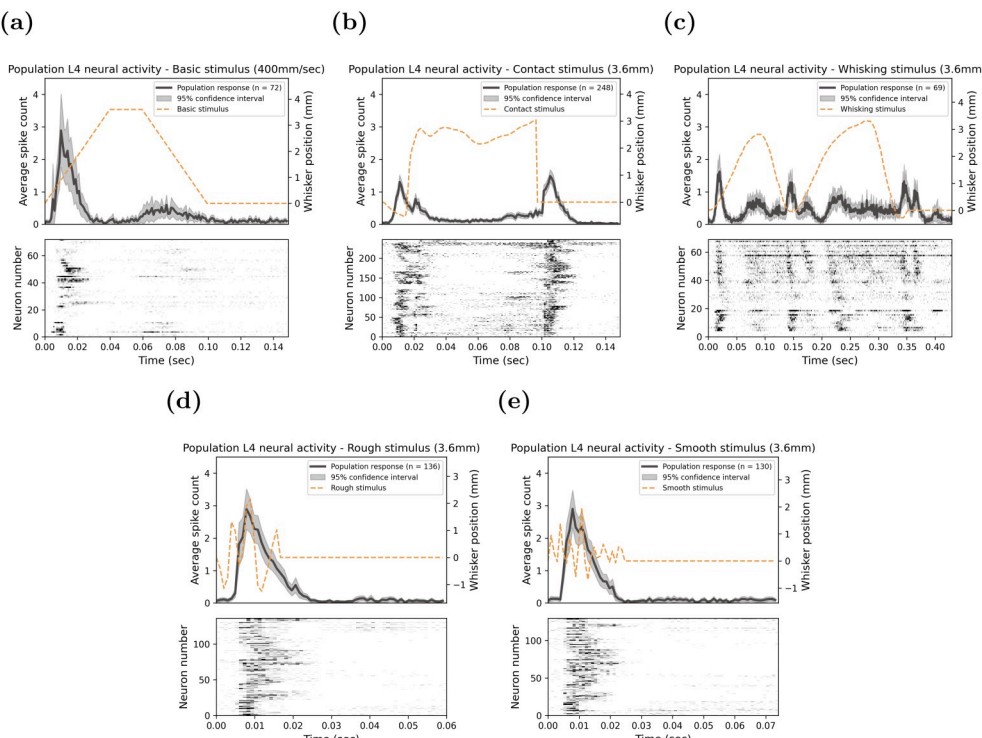

**Fig 2. Layer 4 population response to complex naturalistic stimuli.** Each of these subplots is representative of the strongest whisker deflection of 3.6 mm for each of the following stimuli: **(a)** Basic; **(b)** Contact; **(c)** Whisking; **(d)** Rough; and **(e)** Smooth. The population response measured as the average spike count across all Layer 4 neurons (top panel; grey line) is plotted as a PSTH against the position of the whisker deflection (top panel; yellow dashed line) following the course of the complex naturalistic stimuli. Shaded areas around the population response curve represent the 95% confidence interval. The bottom panel is a raster plot of the individual neurons with the gradient reflecting the intensity of the neuronal response.

where the probability density function, $p(x)$, for single-unit activity vector, $x$, is expressed as a sum over $k$ Gaussian distributions or clusters. $\mu_i$ and $\sigma_i$ are respectively the mean vector and covariance matrix of the $i$th Gaussian distribution, and $\alpha_i$ denotes the probability or the Gaussian weight that the neuronal activity vector $x$ originates from the $i$th Gaussian. The expectation-maximisation (EM) algorithm was used to estimate the model parameters to maximise the posterior probability.

**Optimal number of clusters.** To determine the optimal number of clusters or Gaussian distributions, $k$ (1 to 10), the Bayesian information criterion (BIC) was used to select the appropriate model by balancing between goodness of fit and model complexity. Given that the GMM solutions were sensitive to the model's initial conditions, the GMM was fitted 1000 times with each run initialised by a different seed to obtain a unique set of clustering solutions for that seed. The proportion of cluster agreement, representing a measure of consensus or stability amongst the number of clusters across the 1000 repeated runs, was calculated as follows:

$$\text{Proportion of cluster agreement} = \frac{\text{Frequency of } i \text{ cluster solutions}}{\text{Total number of clustering solutions}}$$

where the frequency of $i$ clusters solution is the count of GMM solutions with $i$ components with the lowest BIC score; and the total number of clustering solutions is the total number of times the GMM was repeated with a different seed, in this case, 1000.

The modal optimal number of clusters with the lowest BIC across the 1000 iterations of different clustering solutions was determined as the optimal number of clusters.

**Model selection.**   Given the GMM clustering algorithm is susceptible to variations in the clustering solutions due to differences in initial conditions, the adjusted Rand index (ARI) was used to assess cluster stability. The ARI statistic measures the degree of agreement between two clustering solutions, corrected for chance using a baseline of expected similarity. It ranges between -1 and 1, where a score of 1 indicates perfect agreement, 0 represents agreement with the expected value, and -1 indicates that the clustering solutions are in complete disagreement with the expected value. The ARI is calculated as follows:

$$\text{Adjusted Rand Index} = \frac{\text{RI} - E(\text{RI})}{\text{Max}(\text{RI}) - E(\text{RI})}$$

where *RI* is the Rand Index (RI), which measures the proportion of neuronal pairs consistently classified in both clustering solutions to the total number of neuronal pairs as defined below; $E(\text{RI})$ is the expected value of the RI given the assumption of random cluster assignment; and Max(RI) is the maximum possible value of the RI, which is equal to 1. The Rand Number is calculated as follows.

$$\text{RI} = \frac{\text{TP} + \text{TN}}{\text{TP} + \text{TN} + \text{FP} + \text{FN}}$$

where TP is the number of true positive neuronal pairs and it refers to the pairs that were identified in the same cluster in both comparative and reference clustering solutions; TP is the number of true negative neuronal pairs and it refers to the pairs that were identified in different clusters in both comparative and reference clustering solutions; FP is the number of false positive neuronal pairs and it refers to the pairs that were identified in the same cluster in the comparative solutions but different in the reference solution; and FN is the number of false negative neuronal pairs and it refers to the pairs that were identified in different clusters in the comparative solutions but in the same cluster in the reference solution.

To determine the seed that produced the most stable clusters, the ARI was calculated by comparing a reference seed against 100 other GMM clustering solutions. The ARI of the reference seed was calculated by averaging the ARI scores across the 100 comparison states, where the average ARI represents the clustering stability of that reference state. This process was repeated another 50 times for each different reference seed and the reference seed with the largest average adjusted ARI was selected as the optimal clustering solution as this indicates the GMM clustering algorithm was able to produce stable clusters that can be reproduced across a large number of repeated clustering solutions.

**Cluster visualisation.**   To visualise the final clustering solution, principal component analysis (PCA) was conducted to linearly transform the single-cell neurons into a new orthogonal basis that maximises the variance of each dimension. Single-unit neurons were distributed against the first two principal components with the largest variance explained and the individual components were plotted to capture the temporal dynamics that explained the largest variance of the neurons in Layer 4.

**Clustering consistency across amplitudes.**   We also examined the consistency of neuronal cluster membership across different stimulus amplitudes of each stimulus, using the same GMM-based approach. For stimuli with more than one cluster, we visualised the clustering patterns by generating a plot of neuron cluster membership as a function of amplitude. In these plots, neurons are represented on the x-axis and amplitudes on the y-axis, with colour coding used to indicate cluster membership and clustering patterns across amplitudes.

Additionally, we computed transition matrices for consecutive amplitude pairs of that stimulus. Each matrix entry $(i, j)$ represents the probability that neurons in Cluster $i$ at amplitude $A_n$ transitioned to Cluster $j$ at amplitude $A_{n+1}$. These matrices were calculated by tracking the cluster membership of neurons at each amplitude, then recording transitions between clusters for consecutive amplitude pairs and finally normalising by the total number of neurons in each cluster at amplitude $A_n$, to obtain transition probabilities.

**Surrogate data set.**   We conducted a permutation test to assess whether the observed clusters are a result of chance using a permutated dataset that had no temporal structure. The timing of the spiking activity of a single unit was shuffled within and between neurons to remove within-neuron and any possible global temporal patterns, respectively. The GMM clustering analysis described above was repeated on the permuted dataset based on the optimal number of clusters determined by the BIC statistic. The BIC for the GMM fitted to the original spike data was compared against 1000 BIC scores derived from fitting the GMM to the permuted spike data. We assessed statistical significance by calculating the number of times the permuted BIC score is less than or equal to the original BIC. This comparison represents the number of times the permutated dataset would outperform the clustering solution of the original dataset, indicating the observed clustering patterns were due to chance and random variability without any underlying patterns or structure in the spike trains.

**Comparison to an alternative clustering method.**   After the optimal number of clusters was identified using the BIC score based on the GMM method, the clustering algorithm was further compared to the Dirichlet Process Mixture Model (DPM). Similar to the GMM, the DPM assigns neurons to clusters by evaluating the statistical similarity of their response patterns based on probabilistic properties. Each neuron is assigned to a cluster according to the likelihood that its temporal response aligns with the mean and variance of that cluster's response. The DPM models each cluster with a Gaussian distribution, defined by a mean vector ($\mu_k$) and a covariance matrix ($\sigma_k$) for the $k$th cluster. The probability that a neuron $x$ belongs to a cluster is represented by the sum of weighted Gaussian distributions:

$$p(x) = \sum_{k=1}^{K} \pi_k \cdot N(x|\mu_k, \sigma_k)$$

The mixing proportions, $\pi_k$, which control the probability of assigning neuron $x$ to the $k$th cluster, are governed by a Dirichlet Process through a stick-breaking process. The stick-breaking process constructs the weights by drawing $\beta_k$ from a Beta distribution as follows:

$$\pi_k = \beta_k \prod_{l=1}^{k-1} (1 - \beta_l), \quad \beta_k \sim \text{Beta}(1, \alpha)$$

where $\alpha$ is the concentration parameter, which plays a role in determining how the clusters are distributed. Specifically, $\alpha$ controls the probability of assigning a neuron to a new cluster. For practical reasons, the model was truncated to a large finite number of components ($K$), approximating the infinite mixture with a sum over $K$ clusters.

## Results

### Identification of temporal response patterns at the highest maximum stimulus amplitude/velocity

Responses from single-units to each of the five stimuli were analysed using a Gaussian Mixture Model (GMM) for clustering to uncover underlying temporal response patterns and groupings of functional sub-populations within Layer 4 of the rat barrel cortex.

**Table 2. The optimal number of clusters identified using the GMM clustering algorithm.** The proportion of cluster agreement is a measure of consensus or stability among the number of clusters across repeated iterations. The total number of clusters was determined by fitting the GMM 1000 times with each run initialised by a different seed to obtain a unique set of clustering solutions for that seed. The Adjusted Rand Index (ARI) evaluates the stability of the clustering solution by measuring the proportion of neuronal pairs consistently classified in both clustering solutions to the total number of neuronal pairs across 100 other solutions.

| Stimulus | Proportion cluster agreement | Total clusters | Adjusted Rand Index |
|---|---|---|---|
| Basic | 1 | 1† | N/A |
| Whisking | 1 | 1† | N/A |
| Contact | 1 | 3*** | 0.877 (0.021) |
| Smooth | 0.92 | 2*** | 0.634 (0.037) |
| Rough | 0.96 | 2*** | 0.692 (0.009) |

*, $p < 0.05$;

**, $p < 0.01$; and

***, $p < 0.001$.

† Statistical significance and ARI were not assessed due to the presence of only 1 cluster each for these stimuli.

The optimal number of distinct temporal response patterns varied among stimuli and, for some stimuli, between different maximum amplitudes of the same stimulus. Table 2 shows the optimal number of clusters for each of the five stimuli at the fastest onset ramp velocity (Basic stimulus) or highest maximum amplitude of 3.6 mm (for the other four other stimuli) and the proportion of cluster agreement, measuring the consistency of the optimal number of clusters across different runs of the clustering analysis. (As described in Methods, the GMM was fitted 1000 times, with each run initialised with a different seed to obtain a unique set of clustering solutions for that seed).

For both the Basic and Whisking stimuli, neuronal responses could be classified into a single consistent temporal response pattern across 1000 different clustering runs. For the Contact stimulus, the single-unit responses were consistently classified into three distinct temporal response patterns in all 1000 clustering runs. The presence of three distinct clusters of temporal response patterns was statistically significant ($p < 0.001$), as indicated by the permutation test, suggesting the non-random formation of these clusters. Finally, the neural responses to both the Smooth and Rough stimuli could be classified into two well-defined temporal response patterns. This classification showed high consistency as demonstrated by a cluster agreement rate of 0.92 and 0.96, respectively. The permutation test confirmed that the total number of clusters for both these stimuli was statistically significant (both $p < 0.001$), greater than what would be expected by chance. Hence, the synchronised single-unit activity likely resulted from the information content of the stimulus input signal. These clusters of temporal response patterns are a consequence of the precise timing and spike count in response to the stimuli.

Once the optimal number of clusters was determined for each stimulus and across the set of maximum amplitudes, we compared the cluster assignments for the same single-unit across different pairs of GMM clustering runs. We used the Adjusted Rand Index (ARI—see Methods) to assess the consistency of cluster membership across the different solutions in the 100 GMM clustering iterations. The ARI index was computed 100 times for each GMM clustering iteration to determine the optimal seed for the Contact, Rough, and Smooth stimuli, which produced two or more clusters of temporal response patterns.

The neural responses to the Contact stimulus demonstrated high similarity and agreement between the different GMM clustering analysis runs. On average, 87.7% of the pairs of single-

unit neuronal responses shared the same cluster assignments across 100 iterations. For the Smooth and Rough stimuli, the agreement between cluster assignments across 100 iterations was, on average, 63.4% and 69.2% of single-unit pairs sharing the same clustering assignments, respectively. Overall, the relatively high ARI for the Contact stimulus suggests that the clustering solutions remained consistent, reproducible, and relatively robust across variations in the initialisation seed of the model. While the robustness of the clustering model was evident for this stimulus, the optimal number of neural response clusters and the consistency of cluster membership across different solutions provide confidence in the identified clusters. However, further analysis is required to fully assess the stability of the clustering solutions across the Rough and Smooth stimuli.

## Characteristics of the neuronal temporal response clusters to the Contact stimulus

As noted above, the GMM identified that, at the maximum value of the primary stimulus variable (onset ramp velocity or highest deflection amplitude), at least three stimuli had more than one type (cluster) of neural temporal response pattern. We then examined the detailed characteristics of the different clusters of temporal response patterns to each of these stimuli, starting with the neural responses to the Contact stimulus for which the GMM had identified three distinct temporal response patterns, as depicted in Fig 3. In the figure, panels a-c present the average temporal response of the single units of the three GMM-defined clusters (a: Cluster 1; b: Cluster 2; c: Cluster 3). In the upper plots of each panel, the Contact stimulus waveform over time is shown as the dashed red line while the average temporal response of the cluster is shown in blue (a), red (b) or purple (c). The responses of the three clusters are directly compared in Fig 3d, with the Contact stimulus waveform again illustrated by the dashed red line.

The majority of the single units (n = 121) were assigned to the Cluster 3 temporal response pattern. This pattern consisted of a relatively small and fast response at the stimulus onset, followed by a comparatively larger response at the stimulus offset, with sparse activity outside these two periods (see Fig 3c). The second most prevalent temporal response pattern, Cluster 1 (n = 77), shown in Fig 3a, displayed a fast and large response at the stimulus onset, followed by sustained activity primarily between 80 ms-100 ms from the stimulus onset, until a rapid offset response. Finally, the least prevalent neural temporal response pattern (n = 49) was that of Cluster 2 (panel c), with broader and larger responses at the onset and offset of the stimulus, but limited sustained activity during the whisker deflection, as shown in Fig 3b.

The lower raster plots in Fig 3a–3c show the average across-trial responses for all units in the cluster, with each row of responses being for an individual single unit. These plots strikingly demonstrate the similarity of single-unit response profiles in any one cluster. This fidelity of single-unit response profiles in each cluster is quantitatively demonstrated in the narrow 95% confidence interval of the cluster's average response in the three upper plots of Fig 3a–3c. Additionally, this fidelity can also be seen in the remarkable similarity of response profiles in the lower-panel raster plots of Fig 3a–3c for individual single-unit responses.

We conducted a Principal Component Analysis (PCA) on the temporal response patterns of the Contact stimulus to identify key components that contribute most significantly to uncover the underlying structure of the temporal response patterns. The outcomes of the PCA are summarised in Fig 3e, where the top three principal components were averaged across the underlying single units belonging to each cluster as determined by the GMM. Although the total variance explained by the top three principal components was not very high at 29.3%, with the top principal component accounting for 12.8% of the overall variance, distinct groupings of single units are evident, as indicated by the colour-coded cluster memberships. As

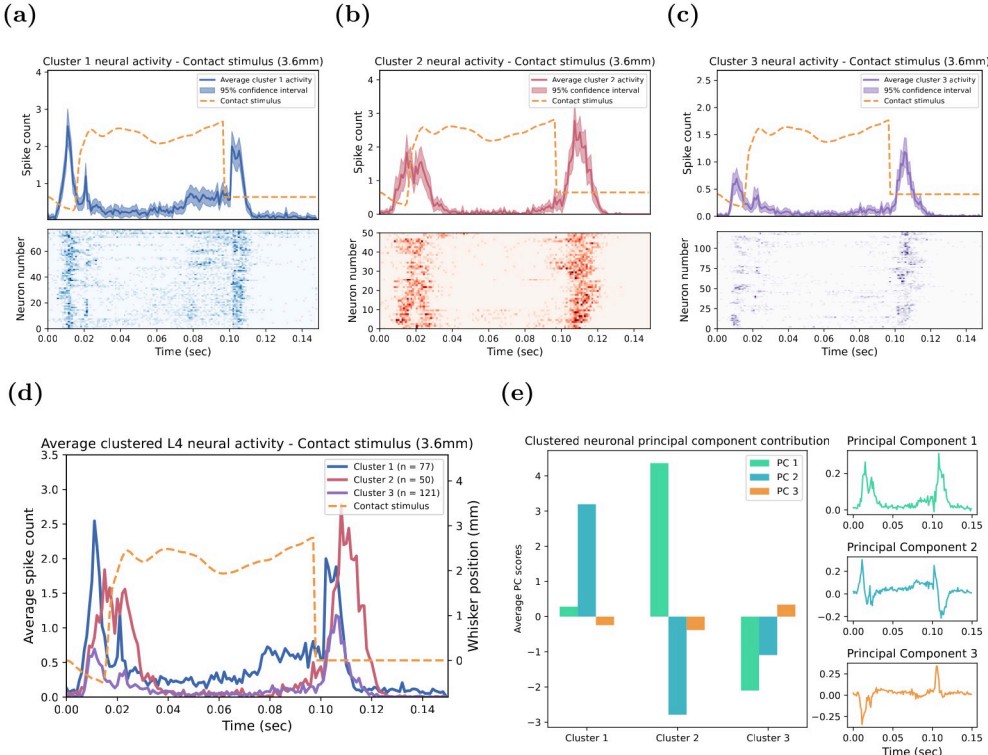

**Fig 3. Characteristics of the temporal response patterns defined by the Gaussian Mixture Model for neuronal responses to the Contact stimulus. (a-c)** Average neural activity for single-unit responses that had the same cluster membership: **(a)** Cluster 1, **(b)** Cluster 2, and **(c)** Cluster 3. The average responses for the three clusters are overlaid in **(d)**. The waveform of the Contact stimulus is the dashed red line in the upper plots of **(a-c)** and in **(d)**. In the upper plots of **(a-c)**, the dark line indicates the average response of Cluster 1 **(a)**, Cluster 2 **(b)**, and Cluster 3 **(c)** respectively, with the shaded region showing the 95% confidence interval of all single-unit responses with the same cluster membership. The bottom panels of **(a-c)** each show a raster plot where each row is the average response of every single unit belonging to that cluster, with the colour gradient indicating the intensity of the neuronal response. **(e)** displays the principal component analysis of L4 neurons, showing the first three principal component averages with the highest eigenvalue for each cluster. The three graphs on the right represent the top principal components over time.

shown in the top of the three vertically-aligned plots to the right in Fig 3e, Principal Component (PC) 1 reveals that the dominant response, accounting for most data variance, stems from two relatively broad excitatory responses at the onset and offset of the Contact stimulus. In contrast, PCs 2 and 3 are made up of distinct fast waveforms, wherein PC 2 displays sharp excitation responses at the onset followed by fast inhibition in the offset, while, conversely, PC 3 demonstrates inhibition at the onset and rapid excitation at the offset of the stimulus.

As mentioned earlier, the total variance explained by the PCA was low, suggesting that the single-unit responses lack dominant linear structures. However, the distinct patterns of excitatory and inhibitory effects of the top principal components may provide further insights into the temporal response patterns defined by the GMM. Indeed, a comparison of the top three principal components against the three clusters of temporal responses revealed some strong associations between the two measures. For instance, PC 1 exhibited similar patterns to the temporal response pattern of single units of Cluster 2 which were characterised by slower and broader onset and offset responses to the stimulus. This association is further supported by the fact that the large positive contribution from PC 1 and the large negative contribution from PC 2 mostly explained the underlying temporal response pattern of single units in Cluster 2, as

shown in Fig 3b and 3e. Correspondingly, the excitatory and inhibitory effects of PC 2 were predominately reflected in Cluster 1. Here, the fast onset excitatory and offset inhibitory activity around 110ms contributed to the fast onset response and offset responses observed in Cluster 1. However, despite the transient inhibitory activity, Cluster 1 neurons exhibited sustained spiking activity throughout the stimulus duration. The inhibitory effect at the offset primarily influences the termination of the response but does not significantly dampen the continuous activity during the stimulus, which accounts for the higher overall activity of Cluster 1 neurons compared to other clusters. In contrast, Cluster 3's weak and fast average response, primarily at the stimulus offset, is largely attributable to the negative contributions from PC 1 and 2, and the positive contribution of PC 3. Specifically, the negative contributions of PC 1 and 2, as well as the positive contribution of PC 3, lead to an overall reduction in the observed activity at the onset of the response. Furthermore, the negative contribution of PC 2 and the slight positive contribution of PC 3 explain the strong and fast offset response of Cluster 3. Overall, the interplay of these three principal components, particularly the first two, appears to play an important role in shaping the dynamics underlying the temporal response patterns of the Contact stimulus determined by the GMM.

## Characteristics of the GMM-defined temporal response pattern clusters for other stimuli

In the same way as detailed above for the Contact stimulus, the characteristics of the GMM-defined single neuronal temporal response pattern for both the Basic and Whisking stimuli are shown in Fig 2a and 2c. This figure illustrates the GMM-defined temporal response patterns for the fastest onset ramp speed for the Basic stimulus and the highest maximum amplitude for the Whisking stimulus. The Basic stimulus evoked a strong response during the stimulus onset ramp, followed by a small response at the offset ramp. Minimal variation was observed across all single units within this singular temporal response cluster, as indicated by the narrow 95% confidence intervals about the average response and the uniformity of the raster plot.

In keeping with the stimulus features, the neuronal responses to the Whisking stimulus exhibited oscillatory response patterns, with the majority of the activity concentrated around the peaks and troughs of the Whisking deflections. Similar to the Basic stimulus, the single-unit responses displayed a high degree of temporal concordance with this stimulus, as further evidenced by the synchronised responses depicted in the 95% confidence interval and the raster plot.

While the overall analysis identified a single dominant temporal response pattern for the Whisking stimulus, we acknowledge that some heterogeneity was observed in the responses, as seen in the 95% confidence interval and the raster plot (Fig 2c). Specifically, there are differences in the responses of certain neurons, such as neurons 1–20 and neurons 40–70, with slight delays in the peaks of activity, particularly at 150 ms, 175 ms, and 225 ms. To further investigate whether reducing the stimulus duration would reveal additional response patterns, we performed additional clustering analysis for the Whisking stimulus with shortened stimulus durations of 150 ms, 175 ms and 225 ms from stimulus onset rather than the full 430 ms stimulus duration. As detailed in Methods, these shorter stimulus durations allowed us to determine whether shorter windows would reveal new clusters with finer temporal dynamics. However, despite this adjustment, the Whisking stimulus continued to evoke only a single temporal response pattern, consistent with the initial analysis performed on the full stimulus duration. This finding, which is illustrated in S1 Fig, suggests that the neuronal response to the Whisking stimulus remains robust and uniform, even when examined over shorter timescales. While

some variability in individual neuron responses was present, this did not result in distinct clusters being identified by the GMM analysis.

Overall, while the Whisking and Basic stimuli evoke a single dominant temporal response pattern, some variability in the responses of specific neurons was observed. Despite these differences, the clustering analysis indicates a high degree of uniformity and synchronicity across most single units, thereby highlighting the robustness of the neuronal responses to these stimuli, even when shorter stimulus durations were examined for the Whisking stimulus. Overall, the single temporal response pattern observed in both the Basic and Whisking stimuli demonstrates a high degree of uniformity and synchronicity across most single units, thereby highlighting the robustness of the neuronal responses to these stimuli.

In contrast to the two stimuli described above, for both the Rough and Smooth stimuli, at the highest maximum amplitude of 3.6 mm, GMM analysis segregated neuronal responses into two clusters of temporal response patterns (Fig 4a and 4d). The temporal response patterns for these two stimuli were very similar: Cluster 1 exhibited a rapid and strong response to the onset of the stimuli, while Cluster 2 displayed a slow and weak response. The temporal response patterns were highly synchronised across single units within the same cluster, as evidenced by the narrow 95% confidence intervals surrounding the average response. This synchronisation was also seen in the raster plots in Fig 4b, 4c, 4e and 4f. The PCA analysis of the neuronal responses to these two stimuli revealed identical underlying patterns (Fig 4g and 4h) based on the top three principal components. For both stimuli, PC 1 had a fast onset response that contributed strongly to Cluster 1 neurons that produced fast and strong responses at the onset of the stimulus. PC 2 consisted of a strong initial inhibition immediately at the onset of the stimulus, followed by slow excitatory for the remaining duration of the stimulus. Therefore, the slow and weak characteristic of Cluster 2's temporal response pattern can be largely attributed to the interplay between the negative influence of both PC 1 and PC 2. Finally, PC 3 displayed a strong mix of rapid and transient inhibition and excitation from the stimulus onset. This component did not contribute to the underlying temporal response patterns for the Rough stimulus, but it did make a minimal contribution to the Smooth stimulus. Similar to the Contact stimulus above, these findings highlight the complex interplay of the principal components in shaping the temporal response patterns of the neuronal clusters.

## Temporal response patterns across variation in the primary stimulus variable for each stimulus

In the sections above, we analysed the neuronal temporal response patterns at the fastest onset ramp velocity for the Basic stimulus and the highest maximum amplitude for all other stimuli. However, each stimulus waveform in our test suite consisted of several stimuli with the same waveform but with variations in the primary stimulus variable of interest in this study. For instance, the Basic stimulus consisted of a suite of five stimuli with trapezoid waveform, each with a different onset ramp velocity. The other stimuli each consisted of a suite of 10 stimuli with the appropriate stimulus waveform but with 10 distinct maximum amplitudes. We now consider the outcome of applying the GMM analysis to all ten different maximum amplitudes for each of the Contact, Rough, Smooth, and Whisking stimuli, and across the five varying onset ramp velocities for the Basic stimulus.

Table 3 summarises the number of GMM-defined clusters for each stimulus across the primary variable of maximum amplitude or onset velocity. The neuronal responses to the Contact stimulus were consistently segregated into three distinct clusters of responses across the ten maximum amplitudes and these cluster solutions were statistically significant ($p < 0.001$) according to the permutation test. For the Rough and Smooth stimuli, the number of distinct

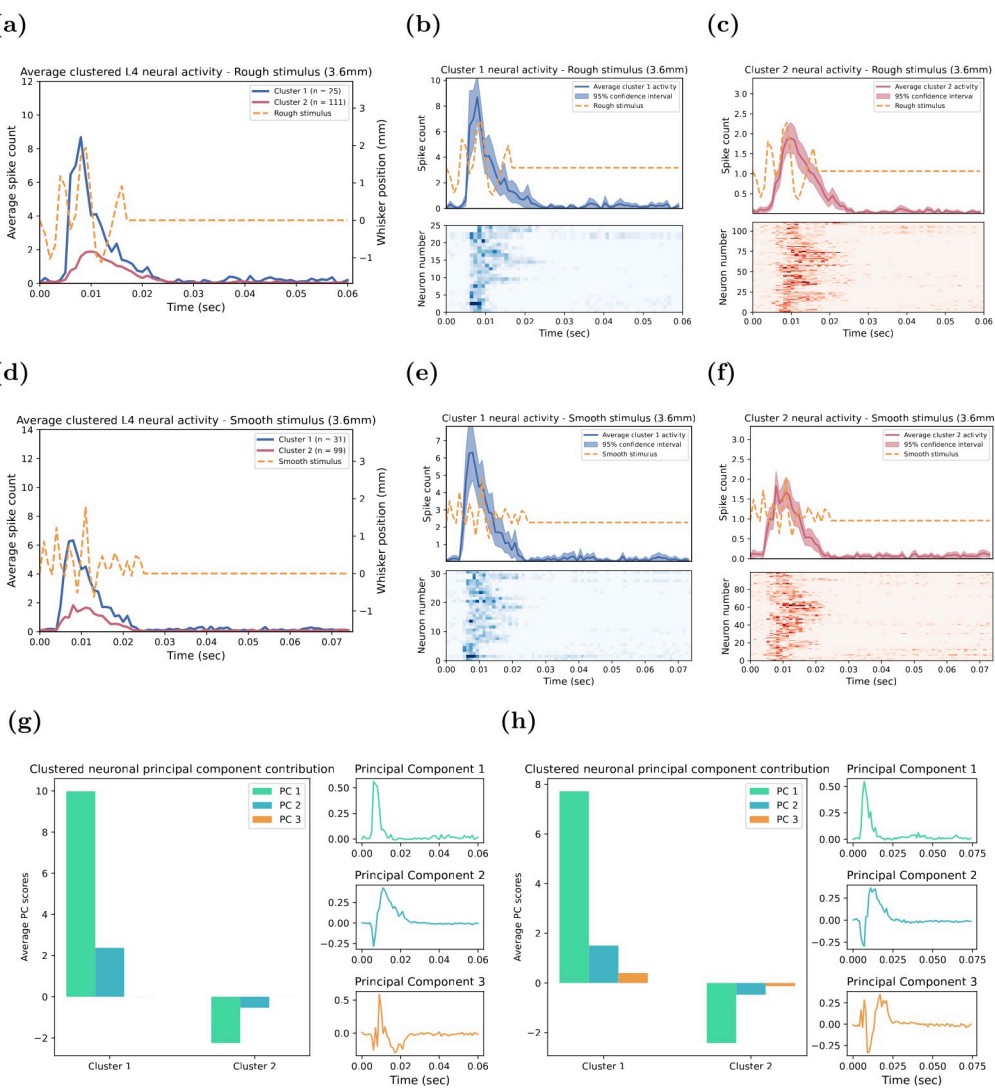

**Fig 4. Comparison of Smooth and Rough stimulus clusters.** Subplot (**a**) and (**d**) show the temporal response patterns of the highest maximum amplitude (3.6 mm) for the Rough and Smooth stimulus analysis, respectively. (**b-c**) and (**e-f**) shows the average neural activity for single-unit responses that had the same cluster membership: (**b-e**) Cluster 1 and (**c**) and (**f**) Cluster 2. The waveform of the Rough and Smooth stimulus is the dashed orange line in the upper plots of (**b-c**) and (**e-f**) and in (**a**) and (**d**). In the upper plots of (**b-c**) and (**e-f**), the dark line indicates the average response of Cluster 1 (**b**) and (**e**) and Cluster 2 (**c**) and (**f**) respectively, with the shaded region showing the 95% confidence interval of all single-unit responses with the same cluster membership. The bottom panels of (**b-c**) and (**e-f**) each show a raster plot where each row is the average response of every single unit belonging to that cluster, with the colour gradient indicating the intensity of the neuronal response. (**e-h**) displays the principal component analysis of L4 neurons, showing the first three principal component averages with the highest eigenvalue for each cluster to the Rough and Smooth stimulus, respectively. The three graphs on the right represent the top principal components over time.

neuronal temporal response patterns defined by the GMM varied among stimuli with different maximum amplitudes. Specifically, for both stimuli, three distinct clusters of neural temporal responses were identified for lower maximum amplitudes of 0.2 mm to approximately 1.6 mm. However, for higher maximum amplitudes, the optimal number of clusters was reduced to two. These clusters for both stimuli were statistically significant across the set of across the set of ten maximum amplitudes, as determined by the permutation test.

**Table 3. Optimal number of clusters identified using the GMM clustering algorithm.** The proportion of cluster agreement represents a measure of consensus or stability amongst the number of clusters across the 1000 repeated runs. The total number of clusters was identified by fitting the GMM 1000 times with each run initialised by a different seed to obtain a unique set of clustering solutions for that seed. The Adjusted Rand Index (ARI) assesses the stability of the clustering solution by measuring the proportion of neuronal pairs consistently classified in both clustering solutions to the total number of neuronal pairs across 100 other solutions.

| Maximum amplitude | Contact | Rough | Smooth |
|---|---|---|---|
| 0.2 mm | 3*** | 3*** | 3*** |
| 0.4 mm | 3*** | 3*** | 3*** |
| 0.8 mm | 3*** | 3*** | 3*** |
| 1.2 mm | 3*** | 3*** | 3*** |
| 1.6 mm | 3*** | 3*** | 2*** |
| 2.0 mm | 3*** | 3*** | 3*** |
| 2.4 mm | 3*** | 2*** | 2*** |
| 2.8 mm | 3*** | 2*** | 2*** |
| 3.2 mm | 3*** | 2*** | 2*** |
| 3.6 mm | 3*** | 2*** | 2*** |

***, $p < 0.001$.

Note: The Basic and Whisking stimuli are not shown in this table as there was only one cluster identified across all maximum velocities and amplitudes, respectively.

When the GMM analysis was applied to the neuronal responses to the Whisking and Basic stimuli, it identified only one cluster of neural temporal response patterns for each of the ten stimuli with varying maximum deflection amplitudes (for the Whisking stimulus) and the five stimuli with different onset ramp velocities (for the Basic stimulus). Specifically, for the Basic stimulus (Fig 5a), there was a monotonic increase in the strength of the neuronal response at stimulus onset as the onset ramp velocity increased. Regarding the timing of the responses, the onset of the neuronal response occurred progressively earlier as the onset ramp velocity increased. Similarly, for the Whisking stimulus (Fig 5b), the single units demonstrated a monotonic increase in response strength as the maximum deflection amplitude increased. In terms of the timing of the responses, the neuronal responses occurred progressively earlier with an increase in the maximum deflection amplitude. Notably, at the lowest maximum amplitudes from 0.8 mm to 1.6 mm, neurons displayed relatively weak onset responses compared to the stronger responses during the rest of the stimulus. This pattern reversed for larger amplitudes from 3.0 mm to 3.6 mm, where the onset response was the strongest response component. Overall, for both the Whisking and Basic stimuli, variations in the maximum deflection amplitudes and onset ramp velocities similarly influenced the characteristics of the strength and timing of neuronal response across both stimuli.

For the Contact stimulus (Fig 5c), neuronal temporal response patterns were highly consistent across the 10 maximum amplitudes of the stimulus. As depicted in Fig 5c, there were consistently three clusters of temporal response patterns, each maintaining approximately the same total number of single units across the range of stimulus maximum amplitudes. Cluster 3 consistently contained the largest number of neurons and was characterised by a dominant pattern that, on average, consisted of weak neuronal responses at the stimulus onset and throughout the stimulus duration, ending in highly concentrated activity at the stimulus offset. Clusters 1 and 2 had relatively similar numbers of single units across the range of stimulus maximum amplitudes and were always fewer than in Cluster 3. Cluster 1 consistently exhibited sharp activity at both the onset and offset of the response across the set of

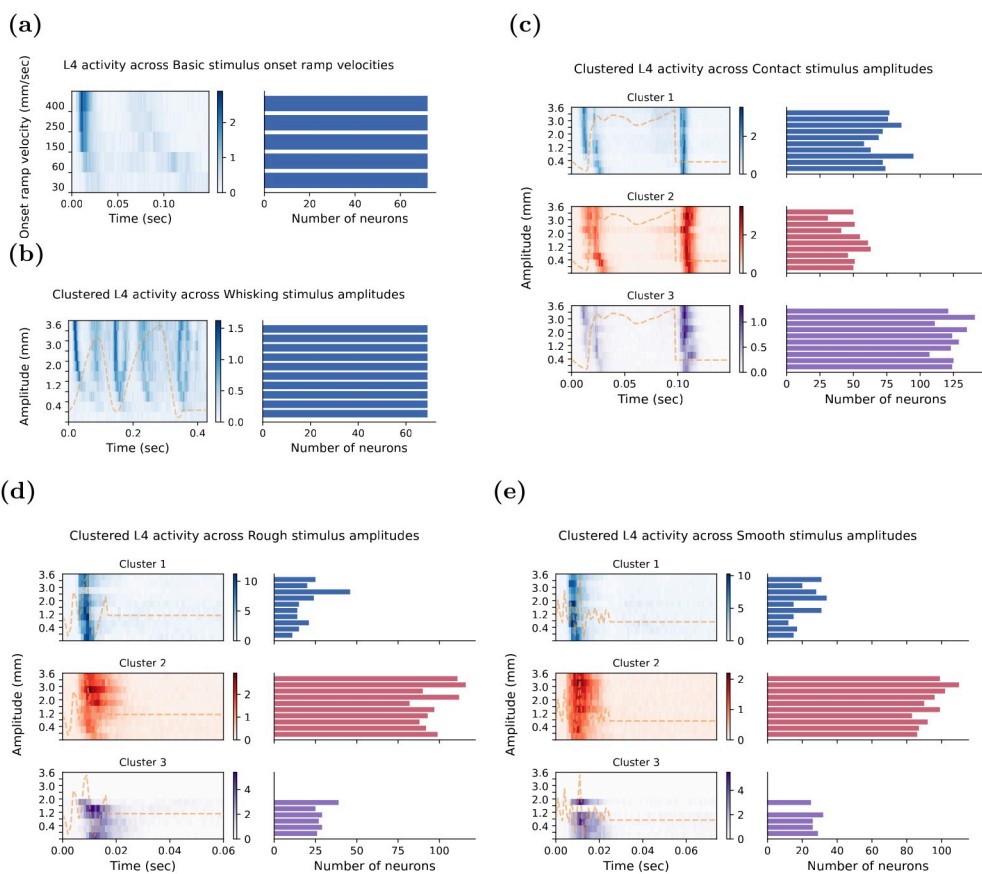

**Fig 5. Temporal response patterns determined by the GMM clustering algorithm across varying primary stimulus parameters (amplitude or onset ramp velocity) of all test stimuli.** Each panel of **(a-e)** shows data for one stimulus. In each panel, the left plot shows the average neuronal activity for each cluster as a heatmap in which the colour intensity indicates the strength of neuronal responses, and the right panel shows as a horizontal bar the number of neurons in that cluster at each test maximum amplitude. The Basic and Whisking responses in **(a)** and **(b)** respectively are shown with only one heatmap since the GMM identified only one cluster of temporal response patterns for each of these stimuli across all variations in the primary stimulus variable (maximum amplitude for the Whisking stimulus; onset ramp velocity for the Basic stimulus). For the Contact **(c)**, Rough **(d)**, and Smooth **(e)** stimuli for which the GMM identified more than one 1 cluster of neuronal responses, the average neuronal response for each cluster is represented in separate plots with Cluster 1 always shown in blue as the top plot; Cluster 2 always shown in red as the second vertical plot, and Cluster 3 always shown in purple as the bottom plot. Note that the heatmaps and the bar plots show how for the Rough **(d)**, and Smooth **(e)** stimuli, three clusters were present at lower amplitudes, but at higher amplitudes only two clusters were present as there were no single units that showed the Cluster 3 response pattern for that stimulus.

amplitudes, with some increasing tonic activity between 80 ms to 100 ms for higher amplitudes, reflecting the slow positive stimulus deflection before offset. In Cluster 2, broad responses in the onset and offset regions of the stimulus were robustly observed, with the onset response becoming increasingly broader temporally at higher amplitudes. For all three clusters, but especially for Clusters 1 and 2, onset neuronal responses occurred progressively earlier as the maximum stimulus amplitude increased. Notably, for the two weakest maximum amplitudes (0.2 mm and 0.4 mm), the onset response occurred at approximately 25 ms, which sharply transitioned to approximately 15 ms for the remaining higher maximum amplitudes. This suggests that the initial negative stimulus deflection was too small to elicit a neuronal response at the lower amplitudes, hence, the onset response seen in the higher amplitudes was

delayed. Finally, for all three clusters, but most clearly evident in Cluster 1, there was a bimodal onset response, which likely represents the initial negative stimulus deflection followed by the immediate positive stimulus deflection.

In general, for the Contact stimulus, the robustness of the three GMM-defined temporal-response-pattern clusters, along with some shared commonality between clusters across stimulus amplitudes, provides some evidence that the stimulus input is driving the generation of these encoded temporal response patterns. To further quantify how neurons behave across amplitudes, we calculated a transition matrix for consecutive amplitude pairs, tracking how neurons switched between clusters as the amplitude changed. A plot of neurons by amplitudes (Fig 6a) shows that neurons largely maintained stable cluster memberships as the amplitudes increased, with only minor transitions observed between clusters. Additionally, the transition matrix (S2 Fig) confirmed that the majority of neurons remained in their original clusters across different amplitudes, with minimal transitions observed between clusters.

Finally, while the GMM identified three clusters of temporal response patterns across the ten maximum amplitudes for both the Rough (**d**) and Smooth (**e**) stimuli, similar to the Contact stimulus, the features of these response patterns differed from those observed for the Contact stimulus, yet were similar to each other. Notably, Cluster 2 emerged as the most prevalent temporal response pattern for both stimuli. The number of neurons in Clusters 1 and 2

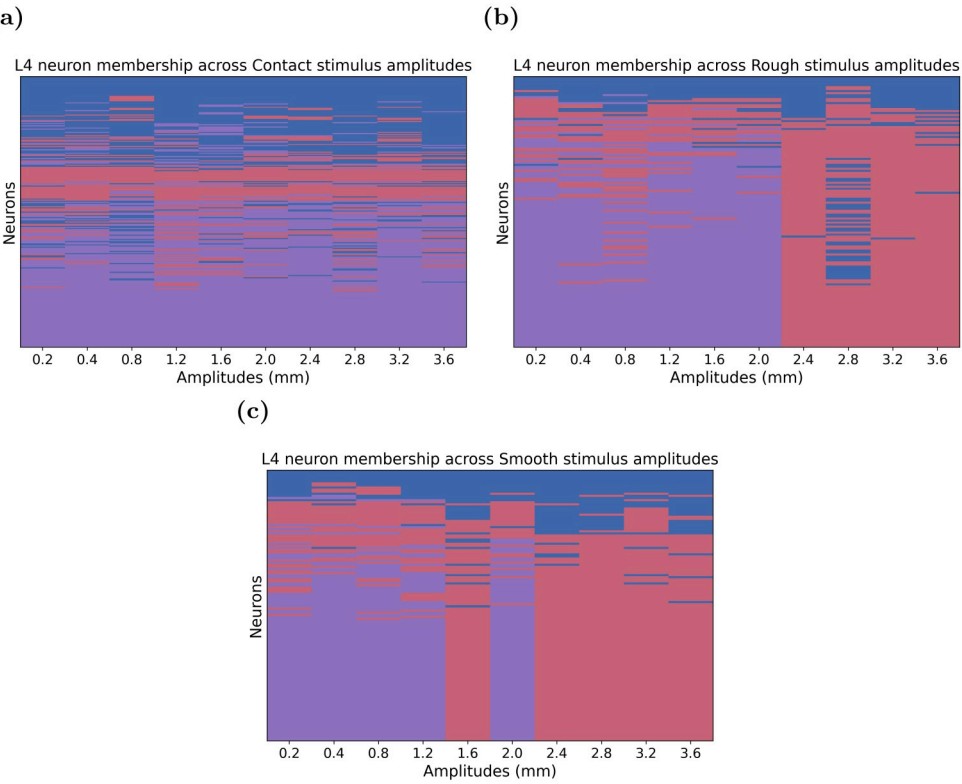

**Fig 6. Neuron membership across maximum amplitudes for the Contact, Rough, and Smooth stimuli.** Each panel shows the cluster membership of neurons across varying maximum amplitudes for the (**a**) Contact stimulus, (**b**) Rough stimulus, and (**c**) Smooth stimulus. In each heatmap, rows represent individual neurons, and columns correspond to the maximum amplitude levels (0.2 mm—3.6 mm). The colours indicate cluster membership, with Cluster 1 in blue, Cluster 2 in red, and Cluster 3 in purple. Neurons that remain within the same cluster across amplitudes maintain consistent colours, while transitions between clusters are marked by changes in colour along the row.

remained stable across the range of maximum deflection amplitude. However, Cluster 3 was only present for low-to-mid amplitudes, during which the number of neurons in Clusters 1 and 3 were similar. Furthermore, Cluster 1 exhibited a slight latency shift with increasing stimulus amplitudes, where the timing of the responses progressively occurred earlier with increasing deflection amplitudes, a pattern similar to what was observed in the other stimuli. Interestingly, Cluster 3 seemed to mirror Cluster 2, as both clusters comprised neurons that responded throughout both types of whisker deflections. However, Cluster 3, on average, displayed larger responses than did Cluster 2. The fact that Cluster 3 only appeared for low-to-mid maximum amplitudes suggests that the GMM may have merged Cluster 3 with Cluster 2 for the higher amplitudes. This is likely because, on average, neuronal responses are stronger and exhibit more consistency at these higher amplitudes, both of which are features of Cluster 2 responses. Overall, this provides a strong indication that neurons in the input Layer 4 encode the Rough and Smooth stimuli using largely identical temporal response patterns.

In the case of the Rough and Smooth stimuli, we observed a dynamic clustering pattern across increasing amplitudes. At lower amplitudes, Clusters 2 and 3 were distinct; however, as the amplitude increased, we observed the merging of Clusters 2 and 3 into a single cluster, Cluster 2. This transition is supported by Fig 6b and 6c, which shows that Cluster 3 disappears for deflection amplitudes greater than 2.4 mm. This is further supported in the transition matrices (S3 and S4 Figs), which show a high probability of neurons from Cluster 3 shifting to Cluster 2 at higher amplitudes. This dynamic shift in clustering across amplitudes aligns with our hypothesis that higher stimulus amplitudes lead to the merging of specific response patterns due to the increased consistency and strength of neuronal responses.

While the above descriptions have focused on the unique characteristics of the clusters for each stimulus, it is also important to note that there were some commonalities shared between clusters of temporal responses to different stimuli. For instance, even though the Basic and Whisking stimuli each had only one cluster, and the Contact, Rough, and Smooth stimuli had multiple clusters, a consistent pattern was observed across all these stimuli. Specifically, the onset of the neuronal responses occurred progressively earlier as the onset ramp velocity (for the Basic stimulus) and maximum stimulus amplitude increased (for the other stimuli). These observations not only provide valuable insights into the differences between various stimuli but also shed light on the general nature of neuronal encoding of sensory information.

## Temporal response patterns classed by the Dirichlet Process

To determine the robustness of different clustering techniques to classify the neuronal temporal response patterns, the GMM-defined temporal response pattern clusters were compared to those identified with the Dirichlet Process Mixture Model (DPM). Here we examined the classification of only the neuronal responses to the Contact, Rough, and Smooth stimuli for each of which the GMM had identified three clusters of neuronal temporal response patterns. This would allow for a more robust comparison than if examining the Basic and Whisking stimuli, for each of which the GMM had identified only a single neuronal temporal response pattern.

Fig 7a–7c provides a summary of the clustering obtained from applying the DPM clustering algorithm to neural responses at the highest maximum amplitude of 3.6 mm of the Contact (Fig 7a), Rough (Fig 7b), and Smooth (Fig 7c) stimuli. These results can be compared to the clustering determined by applying the GMM algorithm to the neuronal temporal response patterns to the same stimuli at the same highest maximum amplitude of 3.6 mm—compare to Fig 3d; Rough stimulus—see Fig 4a; Smooth stimulus—Fig 4d. Such comparison reveals that, overall, the temporal response pattern clusters to the three different stimuli are close to identical between the GMM and the DPM solutions. Remarkably, this is also true when comparing

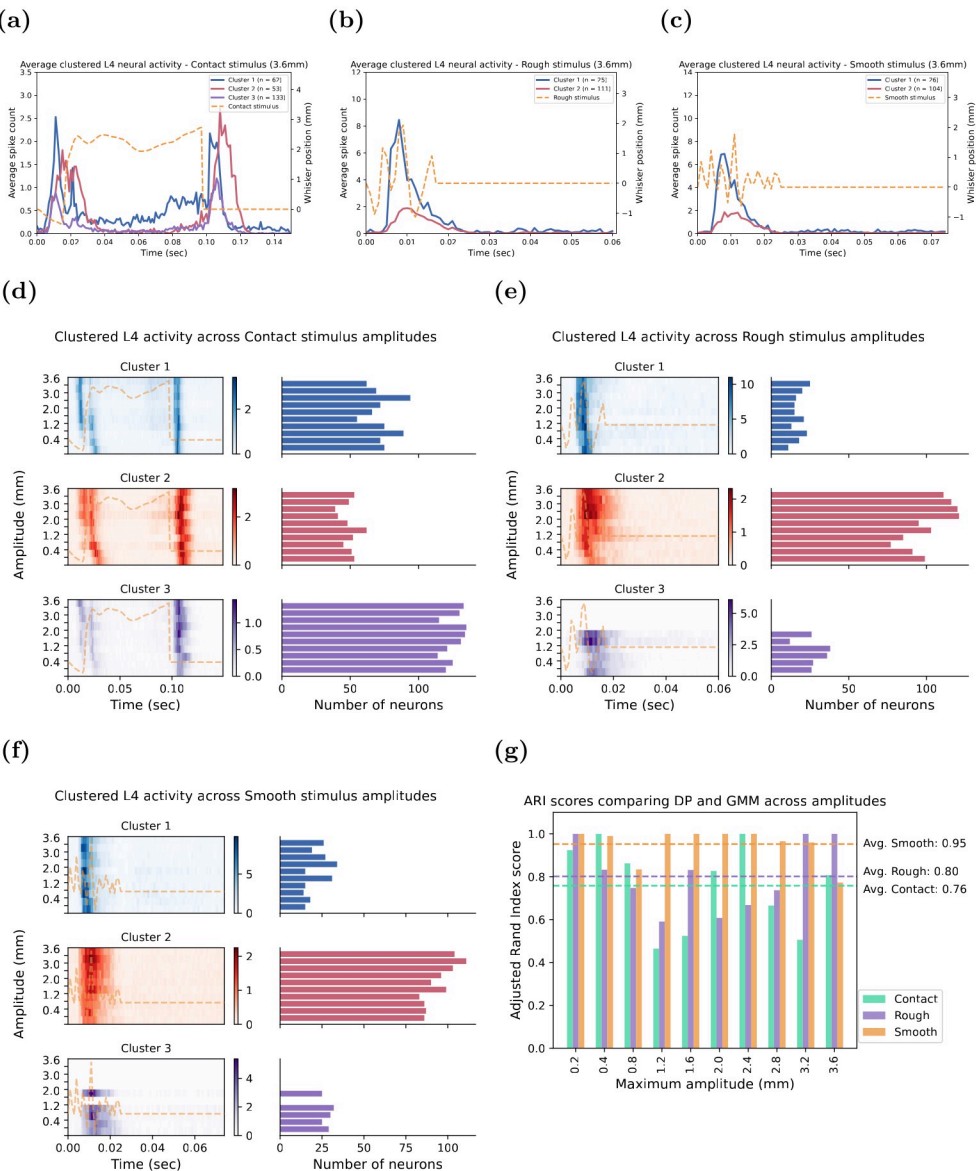

**Fig 7. Temporal response patterns determined by Dirichlet Process-based clustering algorithm across amplitudes and stimuli.** The average neuronal activity for each cluster of the Contact **(a)**, Rough **(b)** and Smooth stimulus **(c)** were determined using the Dirichlet Process Mixture Model (DPM) clustering algorithm. The optimal number of clusters was ascertained from the Gaussian Mixture Model (GMM) analysis to ensure a fair comparison between the two techniques. **(d—e)** shows a heatmap of the average neuronal activity for each cluster with the intensity of the responses measured by the colour gradient. In addition, the number of neurons for each maximum amplitude belonging to the same cluster was illustrated as a horizontal bar chart to the right of the heatmap. The Contact, Rough, and Smooth are in **(d)**, **(e)**, and **(f)**, respectively, with the average neuronal responses of each cluster as Cluster 1 (blue), Cluster 2 (red), and Cluster 3 (purple). **(g)** represents the ARI scores when comparing the cluster membership of single units for the DPM and GMM across maximum amplitudes. The average ARI score across the maximum amplitudes for each stimulus is represented by the colour dashed line.

the temporal response patterns across the suite of ten amplitudes of the Contact stimulus. Specifically, as shown in Fig 7d for the Contact stimulus, the DPM algorithm revealed three distinct patterns, with Cluster 1 showing strong responses concentrated in the stimulus onset and offset periods; Cluster 2 exhibiting broader responses but still concentrated into the onset and

offset periods; and Cluster 3 remaining the most prevalent response that can be characterised as having weaker responses at stimulus onset and throughout the stimulus but highly synchronised in the offset period.

Similarly, for both the Rough (Fig 7b), and Smooth (Fig 7c) stimuli, the DPM algorithm revealed identical temporal response patterns as observed with the GMM solution (viz., Rough stimulus—see Fig 4a; Smooth stimulus—Fig 4d). For both stimuli, Cluster 1 displayed fast and sharp responses to the onset of the stimulus; Cluster 2 was the most prevalent and exhibited slow and relatively weaker responses throughout the stimulus; and Cluster 3 was only present at lower maximum deflection amplitudes and characterised by broad and relatively stronger responses throughout the stimulus. These patterns were identical across all Rough and Smooth stimulus amplitudes and the distribution of the number of single units assigned to each stimulus was relatively similar as shown in Fig 7e and 7f, respectively.

Finally, we compared the cluster membership of each single unit as assigned by the GMM to its DPM assignment, using the ARI to determine the similarity of the two clustering solutions. Fig 7g shows the ARI scores across the maximum amplitudes for the Contact, Rough, and Smooth stimuli. On average, a high degree of cluster membership similarity between the DPM and GMM clustering solution was observed for the single units, where both approaches indicated that the majority of cells belonged to the same cluster. Specifically, the Smooth stimulus had the highest average ARI score of 0.95 across the ten maximum amplitudes, indicating that 95% of the cluster memberships determined by the DPM is in agreement with the membership determined by the GMM. The Rough and Contact also showed strong clustering agreement between the GMM and DPM algorithms, with ARI scores of 0.8 and 0.76, respectively. The consistently strong agreement between the two independent approaches suggests that the clustering solutions were robust and largely captured the underlying temporal response patterns of the neuronal responses. Overall, these clustering similarities and high ARI scores across the Contact, Rough, and Smooth stimuli enhance the reliability and credibility of the GMM-defined clusters.

## Discussion

Our study, which was conducted on a large dataset of Layer 4 single spike-sorted units in the barrel cortex of the halothane-anaesthetized rats, revealed distinct neuronal temporal response patterns. These patterns were specific to the various test stimuli, both naturalistic and artificial. We introduced a GMM-based approach to detect any grouping (clustering) of the temporal response patterns of the single units. This approach also helped us to characterise any changes in the presence of these groups (clusters) across variations in a primary stimulus feature (maximum amplitude for the naturalistic stimuli or onset ramp velocity for the artificial trapezoid stimulus) and between stimuli. Our findings revealed that the different stimuli evoked between 1–3 clusters of temporal response patterns in barrel cortex neurons with the number of clusters specific to particular stimuli and the range of amplitudes/velocities of that stimulus being considered. In cases where there was more than one cluster of temporal response patterns, the clusters were differentiated by the timing and strength of neuronal responses to the stimuli. We applied permutation testing and found that there are consistently three distinct clusters under Contact, Rough and Smooth stimuli at low amplitudes. However, for both the Rough and Smooth stimuli, the number of distinct temporal response patterns decreases to two at higher amplitudes. In these cases, the loss of the third cluster results in a more unified combined temporal response pattern comprising more cells in cluster 1; a response pattern consistent with high (alert) activity at onset (see Fig 5). The robustness of our GMM approach was demonstrated when we undertook an alternative Dirichlet Process-based clustering approach.

The alternative method not only showed that there was agreement with the number of clusters identified by the GMM, but also showed that the temporal characteristics of each cluster remained consistent across different clustering algorithms. This consistency strengthens our confidence in using the GMM method to classify the temporal response patterns of neurons in the input layer of the barrel cortex. We speculate that the formation of these clusters is primarily influenced by the complexity of the stimulus, where this is supported by our observation that the relatively simple Basic and Whisking stimuli, as determined by the NLD coefficient, elicited only one unique temporal response pattern. Furthermore, we propose that the observed temporal response patterns encode specific features about the stimulus which are likely to offer further insights into the specialisation of different clusters of barrel cortex neurons.

### Stimulus complexity and temporal response patterns

Our GMM-based clustering approach revealed that three stimuli, namely the Rough, Smooth, and Contact, each elicited more than a single temporal response pattern in Layer 4 neurons.

In our previous study (Alwis et al., 2012), which employed the same stimuli, we reported that these five stimuli exhibited differences in waveform complexity. This complexity was quantified in that study using the Normalised Length Density measure, where the Whisking and Basic stimuli scored lower on this measure, indicating a lesser degree of complexity compared to the other stimuli, as demonstrated in Table 1. There, it was surmised that this difference in computed waveform complexity was most likely related to the fact that the Whisking and Basic (trapezoid) stimuli possessed fewer distinct whisker deflection features and minimal variation in the overall waveforms. In contrast, the Rough, Smooth, and Contact stimuli exhibited more complex waveforms, leading to a greater number of stimulus components that could likely trigger novel responses in whisker pathway neurons.

In the samples we studied, we observed that signals with lower structural complexity (Whisking and Basic) resulted in reduced diversity in neuronal temporal response patterns. This observation suggests that a simpler waveform might not necessitate the activation of the full suite of inhibitory and modulatory interactions that shape neuronal response patterns in sensory feature detection. For the Basic stimulus, encoding of this feature may be all that is required to signal changes in the dominant whisker-activating feature of this stimulus, namely the velocity of the onset ramp. Correspondingly, we observed that most neuronal activity within the population response is concentrated during periods with high velocities, such as the onset and offset ramps of the stimulus (Fig 5a). These high-velocity periods seem sufficient to generate a consistent temporal response pattern across Layer 4 neurons, with minimal variations. Such velocity-driven responses are consistent with the way primary afferent neurons in the rat's whisker pad deflections, where velocity is a dominant factor in driving activity and this information is preserved as it travels from the whisker pad up to Layer 4 of the barrel cortex [23]. Extending this argument to the Whisking stimulus, the dominant whisker-activating features for this stimulus are also likely to be the velocities or changes in velocities at key points within each cycle, specifically, at the onset, peak, and trough of the stimulus that can be modelled simply by a sinusoidal waveform. Correspondingly, the neuronal population responding to the Whisking stimulus showed one cluster of temporal response patterns, in which all single units showed synchronous activity at three time points within each cycle of the roughly sinusoidal stimulus.

It is important to consider that the extended duration of the Whisking stimulus (430 ms) may contribute to temporal averaging, potentially masking variations in neuronal response patterns that could become apparent over shorter time scales. To explore this, we conducted

additional clustering analyses using reduced stimulus duration of 150 ms, 175 ms and 225 ms. Despite this adjustment, no new clusters were identified, suggesting that the complexity of the stimulus, rather than its duration, plays a more significant role in driving the formation of distinct temporal response patterns in complex stimuli. Whether this finding applies to other stimuli and cortical layers remains an open question. In contrast, the Rough, Smooth, and Contact stimuli are relatively more complex, as evidenced by visual inspection of their waveforms and confirmed by their NLD scores. The increased complexity of these stimuli is likely attributed to the larger number of deflection amplitude variations that occur throughout the stimulus duration, coupled with the frequency of whisker direction changes. These higher-order complexity stimulus features are likely the drivers behind generating more than one distinct neuronal temporal response pattern for these three stimuli. The temporal compactness of the Rough, Smooth, and Contact stimuli, with periods of high velocities closer together, may further explain the emergence of multiple clusters. In comparison, the Whisking and Basic stimuli, with longer and more uniform waveforms, might not recruit the same diversity of neural circuits. While this study did not test the same stimulus at different time scales, future investigations could examine whether applying the same stimulus at different temporal resolutions would reveal additional clusters or distinct neural populations. It must also be noted that while the number of different clusters of neuronal temporal response patterns was always more than one for the Rough, Smooth and Contact stimuli, for some of them, the exact number of clusters also varied with maximum stimulus amplitude (discussed in the next section below). This variation suggests that the amplitude of individual whisker-activating features also plays an important role in triggering these responses. These considerations indicate that these neurons encode various whisker-activating features to sufficiently build representation related to stimulus characteristics such as surface texture, shape, and location.

## Stimulus encoding of clusters of temporal responses

In our analysis of neuronal responses to various whisker deflection stimuli, we identified distinct temporal response patterns for the Contact, Rough, and Smooth stimuli as we have shown in Figs 3 and 4. For the Contact stimulus, we categorised these patterns into three broad groups based on their speed, timing and duration of the response. Generally, we found only a small number of clusters, usually 2 or 3, of distinct response patterns with some minor variation around these clusters.

**Phasic on-off response.** The first group/cluster of neuron temporal response pattern is what we characterised as the *phasic on-off response*, where the underlying single neurons for this cluster responded strongly and rapidly following sudden changes or transitions in the stimulus. For example, in the Contact stimulus, we showed that Cluster 3 (Fig 3c) exhibited a rapid burst of activity at the stimulus onset and offset, immediately followed by a sudden decrease in the response with minimal activity outside these two time periods. Notably, this phasic on-off response was robust across the stimulus amplitudes. However, at weaker maximum stimulus amplitudes, the initial negative deflection was not strong enough to trigger a response from these neurons. Instead, we saw the first response occurring at approximately 25 ms, coinciding with the larger second deflection at 10 ms, as seen in Fig 5c. Interestingly, as the maximum amplitude increased, the later response occurring around 25 ms monotonically decreased, leading to a gradual shift in the earlier response. This dynamical shift in the neuronal activity shows that this cluster primarily responds to the first whisker deflection that exceeds a certain sub-threshold required to elicit a response. Overall, we interpret these phasic on-off neurons as likely to encode information about the velocity and acceleration of the whisker deflection. These neurons would be particularly

relevant for tasks that require rapid processing, such as collision detection or tracking moving objects in the environment, with the common theme that this cluster can rapidly respond to dynamic sensory changes.

**Prolonged on-off response.** The second group of neurons displayed *prolonged on-off response*, where these neurons fired for an extended period during the onset and offset of the stimulus. This response is seen in Cluster 2 (Fig 3b) of the Contact stimulus, which showed slower and broader neuronal responses in the onset and offset period of stimulus, lacking the rapidity of the phasic component seen in Cluster 3. We believe these neurons play a distinct role in encoding continuous information about the position and direction of the whisker deflection. Unlike the phasic on-off neurons, this group displayed slower delayed responses that were likely a result of the integration of the sequence of whisker movements, enabling the neurons to encode specific contextual information about the whisker motion. This integration resulting in slightly delayed neuronal responses would primarily involve neurons and neural circuits to combine information from other neurons to create a meaningful representation of the Contact stimulus. Therefore, these prolonged on-off neurons are likely crucial for encoding and integrating sensory input to provide meaningful information about the stimulus, including surface texture discrimination and object recognition.

**Tonic response.** The final group of neurons we observed were the *tonic neurons*, which exhibited sustained activity throughout the stimulus. Unlike the on-off responses, we found these neurons maintained their activity throughout the entirety of the stimulus duration as long as the stimulus was present. For instance, Cluster 1 (Fig 3a) of the Contact stimulus displayed phasic on-off-like responses but was largely differentiated by the continuous spiking activity throughout the entire stimulus. We believe the neurons underlying this cluster are important for encoding continuous, uninterrupted sensory information to enable continuous monitoring of the stimulus. This cluster likely played an important role in signalling to the brain that the whisker is still maintaining contact with the object throughout the entire Contact stimulus. The rapid onset and offset response periods are likely an important feature of the monitoring activity as they would enable neurons to encode sudden changes in whisker position direction.

In our clustering analysis of the Rough and Smooth stimuli, we observed a small subset of the neuronal population evoked a response similar to the phasic and prolonged on-off responses seen in the Contact stimulus. Neurons underlying Cluster 1 (Fig 4b and 4e) evoked an immediate response at the onset of the stimulus followed by a rapid decline in the average spike count. Interestingly, we did not observe a response at the offset of the stimulus. This absence could be due to the short duration of the Smooth and Rough stimuli, which might not provide adequate time for neurons to differentiate between the onset and offset components of the stimulus. Furthermore, the relatively weak whisker deflection at the offset of the stimulus is unlikely to exceed the sub-threshold necessary to elicit a neuronal response, as observed in the lower maximum amplitudes of the Contact stimulus. Therefore, we believe the likely primary function of Cluster 1 is to detect the stimulus onset due to its rapid response during this period.

In contrast, we found that the large majority of the neurons exhibited a prolonged response at the onset of the stimulus, which was sustained until the end of the stimulus. Unlike the phasic response observed in Cluster 1, neurons in Cluster 2 (Fig 6c and 6f) displayed a prolonged response and are more likely to encode contextual information about the stimulus given the sustained activity. Despite Cluster 2 not demonstrating a significant discernible difference between the Rough and Smooth stimuli, our unpublished analysis into other layers of barrel cortex does show a markedly distinct temporal response pattern between the two stimuli, primarily in the output Layer 5. This difference likely results from the processed and integrated

nature of the information at this level, outlining the hierarchical processing of sensory information within the barrel cortex.

The absence of distinct neuronal populations for the Basic and Whisking stimuli, despite their observed variability, may be attributed to the relatively simple and uniform structure of these stimuli. Unlike the Rough and Smooth stimuli, which exhibit more complex deflections and higher variability in whisker movement, the Basic and Whisking stimuli may not engage a broad range of sensory processing mechanisms. The longer duration of the Whisking stimulus (430 ms) may also contribute to temporal averaging, potentially masking subtle differences in neuronal response patterns. While we examined shorter stimulus durations to uncover possible hidden clusters, no additional clusters emerged, indicating that stimulus complexity, rather than duration, plays a more prominent role in driving diverse temporal response patterns.

In stimuli where periods of high velocities are temporally compact, such as Rough and Smooth, we observed multiple clusters, suggesting that more complex stimuli recruit distinct neuronal populations. In contrast, the Basic and Whisking stimuli, with fewer changes in whisker velocity, are less likely to trigger diverse responses. This highlights the importance of temporal compactness in generating distinct neuronal activity patterns, likely driven by rapid shifts in sensory input. These findings raise the question of whether the differences in the number of clusters produced by different stimuli are due to variations in the circuits recruited or the intrinsic electrical properties of the neurons. Future studies examining different temporal scales or introducing more variable stimulus features could help disentangle how these factors contribute to the observed clustering patterns. A limitation of our study is the independent nature of the recordings, which makes it difficult to assess the coherence of cluster membership across different stimuli, such as Rough and Smooth. As neurons tested under each stimulus were not necessarily the same, it is challenging to reliably track whether individual neurons maintained their cluster membership across these stimuli. Future research that involves recording from the same neuronal population under different stimuli would provide valuable insights into the consistency of temporal response patterns across varying sensory conditions.

## Conclusion

Our investigation of Layer 4 single spike-sorted units in the rat barrel cortex has uncovered diverse temporal response patterns across stimuli. The introduction of a GMM-based clustering approach, complemented by robustness testing through the DPM and permutation testing, demonstrated the reproducible and stimulus-driven nature of identified clusters. The differentiation of neuronal responses by speed and strength in situations with more than two clusters suggests a relationship between stimulus characteristics in shaping neuronal responses. Notably, the distinct temporal response patterns are likely to encode specific features about the stimulus, providing valuable insights into the specialisation of neurons in the barrel cortex. However, it is likely these temporal response patterns are unique to neurons within Layer 4 given it is the first layer of the neocortical circuitry that encodes information about the stimulus. This warrants future research to extend this analysis to consider interactions between Layer 4 and other cortical layers, as exploring the hierarchical relationships and information flow among different layers may reveal insights into how sensory information is integrated and transformed across cortical circuits. Further studies may also consider the robustness of these identified clusters to injured conditions, such as traumatic brain injury (TBI) and stroke, as this could provide insights into the changes of specific encoding mechanisms that may be impacted under conditions of injury.

## Supporting information

**S1 Fig. Clustered neural activity during Whisking stimulus across shortened analysis windows.** This figure shows the clustering analysis for clustered neural activity in response to the Whisking stimulus at the highest maximum amplitude (3.6 mm) for three different time windows: **(a)** 150 ms, **(b)** 175 ms, and **(c)** 225 ms from stimulus onset. In each panel, the top plot represents the average spike count for neurons in Cluster 1 (solid blue line) along with the 95% confidence interval (shaded blue region), and the Whisking stimulus waveform is overlaid (dashed orange line) for reference. The bottom plot shows the corresponding raster plot of neural activity across the population of neurons in Cluster 1, where each row represents a single neuron.
(TIF)

**S2 Fig. Transition matrices across maximum amplitudes for the Contact stimulus.** Each heatmap shows the transition probabilities of neurons between clusters across consecutive amplitude levels for the Contact stimulus. The x-axis represents the cluster assignments at the next amplitude, while the y-axis represents the cluster assignments at the previous amplitude. Colours indicate the probability of neurons transitioning between clusters, with darker shades representing higher probabilities. Each transition matrix illustrates how neurons in Cluster 1, Cluster 2, and Cluster 3 change or maintain their cluster membership as the stimulus amplitude increases from one level to the next (0.2 mm → 0.4 mm, 0.4 mm → 0.8 mm, etc.).
(TIF)

**S3 Fig. Transition matrices across maximum amplitudes for the Rough stimulus.** Each heatmap shows the transition probabilities of neurons between clusters across consecutive amplitude levels for the Rough stimulus. The x-axis represents the cluster assignments at the next amplitude, while the y-axis represents the cluster assignments at the previous amplitude. Colours indicate the probability of neurons transitioning between clusters, with darker shades representing higher probabilities. Each transition matrix illustrates how neurons in Cluster 1, Cluster 2, and Cluster 3 change or maintain their cluster membership as the stimulus amplitude increases from one level to the next (0.2 mm → 0.4 mm, 0.4 mm → 0.8 mm, etc.).
(TIF)

**S4 Fig. Transition matrices across maximum amplitudes for the Smooth stimulus.** Each heatmap shows the transition probabilities of neurons between clusters across consecutive amplitude levels for the Smooth stimulus. The x-axis represents the cluster assignments at the next amplitude, while the y-axis represents the cluster assignments at the previous amplitude. Colours indicate the probability of neurons transitioning between clusters, with darker shades representing higher probabilities. Each transition matrix illustrates how neurons in Cluster 1, Cluster 2, and Cluster 3 change or maintain their cluster membership as the stimulus amplitude increases from one level to the next (0.2 mm → 0.4 mm, 0.4 mm → 0.8 mm, etc.).
(TIF)

## Acknowledgments

We would like to acknowledge Dr Duwage Alwis, Dr Ben Allitt, and Dr Kate Gillespie-Jones for providing the original datasets used in this study.

## Author Contributions

**Conceptualization:** Dinh K. Tang, Mark B. Flegg, Ramesh Rajan.

**Data curation:** Dinh K. Tang.

**Formal analysis:** Dinh K. Tang, Mark B. Flegg.

**Investigation:** Ramesh Rajan.

**Methodology:** Dinh K. Tang, Mark B. Flegg, Ramesh Rajan.

**Project administration:** Ramesh Rajan.

**Software:** Dinh K. Tang.

**Supervision:** Mark B. Flegg.

**Validation:** Dinh K. Tang, Mark B. Flegg.

**Visualization:** Dinh K. Tang.

**Writing – original draft:** Dinh K. Tang.

**Writing – review & editing:** Dinh K. Tang, Mark B. Flegg, Ramesh Rajan.

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
