## [Decision Letter · Decision Letter 0]

17 Jul 2024

PONE-D-24-22742Temporal Response Patterns of Layer 4 Rat Barrel Cortex Neurons Across Various Naturalistic Whisker MotionsPLOS ONE

Dear Dr. rajan,

Thank you for submitting your manuscript to PLOS ONE. After careful consideration, we feel that it has merit but does not fully meet PLOS ONE’s publication criteria as it currently stands. Therefore, we invite you to submit a revised version of the manuscript that addresses all the points raised during the review process.

We look forward to receiving your revised manuscript.

Kind regards,

Stéphane Charpier

Academic Editor

PLOS ONE

Journal Requirements:

3. In the online submission form, you indicated that The data underlying the results presented in the study are available from the authors.

Reviewers' comments:

Reviewer's Responses to Questions

**Comments to the Author**

1. Is the manuscript technically sound, and do the data support the conclusions?

Reviewer #1: Yes

2. Has the statistical analysis been performed appropriately and rigorously? 

Reviewer #1: Yes

3. Have the authors made all data underlying the findings in their manuscript fully available?

Reviewer #1: Yes

4. Is the manuscript presented in an intelligible fashion and written in standard English?

Reviewer #1: Yes

5. Review Comments to the Author

**Reviewer #1:** The authors here study the temporal patterns of single unit responses to basic whisker deflections and to naturalistic stimuli. Using a Gaussian Mixture Model and then the Dirichlet Process to group neurons into clusters, they find multiple clusters of neurons using either a contact-like, and smooth texture or a rough texture-like stimulus. Simple stimuli (as indicated by their low normalized length density), such as a single whisker deflection or whisking, never segregated neurons, regardless of the amplitude or the velocity used. In contrast, complex stimuli such as those mimicking contact with an object or brushing against a smooth or rough texture segregated neurons into 2 to 3 clusters. Interestingly, rough and smooth textures produced clusters with similar temporal patterns: rapid and strong responses in one cluster, and slow and weak responses in the other cluster. The stimulus that mimicked contact with an object produced three different temporal patterns: phasic or prolonged on-off responses and tonic responses.

Major:

While I find the study interesting, it would be greatly improved if information about the coherence of clusters between stimuli were provided, where it’s possible. Indeed, the numbers of rats and cells reported in Table 1 suggest that “rough” and “smooth” were tested on the same neurons. If this is correct, and because the temporal profiles obtained are similar and the number of clusters is identical, with the same dependence on amplitude, it would be informative to know whether a neuron was in, say, cluster 1 for both “rough” and “smooth”. This analysis would also strengthen the GMM clustering that has the lowest Adjusted Rand Index for these stimuli (table 2).

Similarly, since different amplitudes were used, for “contact”, “rough” and “smooth” , it would also improve the study to provide such “consistency index” indicating the extent to which neurons were classified in the same cluster, across amplitudes. In the case of rough and smooth, it would also address the authors hypothesis that clusters 3 and 2 merge when stimuli of large amplitudes are used (line 566).

Also, separation of putative interneurons from excitatory cells would be a very good addition (give proportions for each cluster?).

Finally, the fact that GMM did not reveal different populations of neurons for the “basic” and “whisking” stimuli is intriguing. For example, it is difficult to understand why the phasic and prolonged responses are not apparent. What might be the confounding factors? This should be discussed because the patterns obtained with “whisking” do indeed show some variability. This is visible in the 95% confidence interval, which is quite large between peaks, and in the raster plot (Fig 2c, Fig. 5d). In particular the responses of neurons 1 to 20 seem different from the rest. Beside, for neurons 40-70, one can see some delays in the peaks of activity that are maintained throughout the two cycles of the “whisking” stimulus (times 0.150, 0.175, 0.225 sec…). Could it affect the ability to find clusters?

Related to the question above, is it possible that multiple clusters can only be found when periods of high velocities are close from each other in the stimulus? Indeed, we see that stimuli that are complex, that produce multiple clusters, are in fact stimuli that are temporally compact (Table 1). It is unfortunate that the same stimulus was not applied at two different time scales (25 ms and 340 ms for example). This would have added to the question of whether the differences in the number of clusters produced by the different stimuli are due to differences in the circuits that are recruited, as proposed in the discussion (lines 697-701), or due to the intrinsic electric properties of the cells, which could also be mentioned in the discussion.

Minor:

Line 55: we have identified that the formation of temporal response patterns is mainly dependent on stimulus complexity and the strength and speed of the whisker deflection.

I understand the statement about complexity and strength, but not speed: changing the velocity of the basic stimulus did not allow patterns of responses to be separated.

Line 71: A sequence of PW deflections was conducted using each of the five stimulus waveforms depicted in Fig 1 .

This sentence suggests that multiple stimuli were tested in each recording. Please, correct if it is not the case.

Line 82-85: Please, indicate the grain corresponding to rough and smooth

Line 141 Change panel a for b.

Line 149-150. Panels should go from a to e, not a to d. Panel a is for “basic”

Fig. 3. I do not think this figure is necessary

Line 245: Please, describe the method or refer to a paper describing the Dirichlet Process.

Line 290-297. Overall, the high ARI for the neural responses to these three stimuli suggests that the clustering solutions remained highly consistent, reproducible, and relatively robust across variations in the initialization seed of the model. The robustness in the optimal number of neural response clusters and in the cluster membership across different clustering solutions provides strong confidence that the identified clusters reliably represent the underlying neuronal temporal response patterns.

I would tone down that sentence a bit given that the adjusted Rand Index was high for only 1 of the 3 stimuli that produced clusters.

Line 373 Correspondingly, the excitatory and inhibitory effects of PC 2 were predominately reflected in Cluster 1. Here, the fast onset excitatory and offset inhibitory activity around 110ms contributed to the fast onset response and offset responses observed in Cluster 1.

It is unclear why we do not see more of the inhibitory activity in the averaged PSTH of cluster 1 neurons. In fact, Cluster 1 neurons appear to be more active than other neurons. They are later described as “Tonic” neurons. Isn’t there a contradiction here?

Fig. 5 and lines 388-417; Are these necessary since it was said earlier that GMM yielded a single cluster for “basic” and “whisking” stimuli? The only new information in Fig. 5, namely the 95% confidence interval, could already be shown in Fig. 2, for all stimuli.

Line 416: Overall, the single temporal response pattern observed in both the Basic and Whisking stimuli demonstrates a high degree of uniformity and synchronicity across all single units, thereby highlighting the robustness of the neuronal responses to these stimuli.

There is heterogeneity in the responses to whisking, as visible in the 95% confidence interval and in the raster plot of Fig. 5d and Fig. 2c. In particular the responses from neurons 1 to 20 appear to be different from the rest. Also, one can see some delays in the peaks of activity between neurons 40-70 that are maintained throughout the whisking stimulus (times 0.150, 0.175, 0.225 sec).

Line 701: Our observations suggest that this phenomenon is not confined to the cortex and could be present at lower levels of the whisker pathway.

What observations?

6. PLOS authors have the option to publish the peer review history of their article (what does this mean?). If published, this will include your full peer review and any attached files.

Reviewer #1: No

---

## [Author Response · Author response to Decision Letter 0]

25 Nov 2024

Dear Charpier

Thank you for your valuable constructive comments and suggestions that have helped improve the quality of our manuscript. We have carefully addressed all of the points raised during the review process.

As requested, we have included within this resubmission:

- A rebuttal letter detailing our responses to each comment; 

- A marked-up version of the manuscript with track changes; and

- An unmarked version of the revised manuscript.

We appreciate your consideration and look forward to your feedback.

Kind regards

Ramesh

---

## [Decision Letter · Decision Letter 1]

3 Dec 2024

Temporal Response Patterns of Layer 4 Rat Barrel Cortex Neurons Across Various Naturalistic Whisker Motions

PONE-D-24-22742R1

Dear Dr. rajan,

We’re pleased to inform you that your manuscript has been judged scientifically suitable for publication and will be formally accepted for publication once it meets all outstanding technical requirements.

Kind regards,

Stéphane Charpier

Academic Editor

PLOS ONE

Additional Editor Comments (optional):

Reviewers' comments:

Reviewer's Responses to Questions

**Comments to the Author**

1. If the authors have adequately addressed your comments raised in a previous round of review and you feel that this manuscript is now acceptable for publication, you may indicate that here to bypass the “Comments to the Author” section, enter your conflict of interest statement in the “Confidential to Editor” section, and submit your "Accept" recommendation.

Reviewer #1: All comments have been addressed

2. Is the manuscript technically sound, and do the data support the conclusions?

Reviewer #1: Yes

3. Has the statistical analysis been performed appropriately and rigorously? 

Reviewer #1: Yes

4. Have the authors made all data underlying the findings in their manuscript fully available?

Reviewer #1: Yes

5. Is the manuscript presented in an intelligible fashion and written in standard English?

Reviewer #1: Yes

6. Review Comments to the Author

Reviewer #1: (No Response)

7. PLOS authors have the option to publish the peer review history of their article (what does this mean?). If published, this will include your full peer review and any attached files.

Reviewer #1: No

---

## [Editor Report · Acceptance letter]

8 Dec 2024

PONE-D-24-22742R1 

PLOS ONE

Dear Dr. Rajan, 

I'm pleased to inform you that your manuscript has been deemed suitable for publication in PLOS ONE. Congratulations! Your manuscript is now being handed over to our production team.

Kind regards, 

on behalf of

Pr. Stéphane Charpier 

Academic Editor

PLOS ONE